# Peering into lunar permanently shadowed regions with deep learning

V. T. Bickel [1✉], B. Moseley [2], I. Lopez-Francos [3] & M. Shirley [3]

The lunar permanently shadowed regions (PSRs) are expected to host large quantities of water-ice, which are key for sustainable exploration of the Moon and beyond. In the near future, NASA and other entities plan to send rovers and humans to characterize water-ice within PSRs. However, there exists only limited information about the small-scale geomorphology and distribution of ice within PSRs because the orbital imagery captured to date lacks sufficient resolution and/or signal. In this paper, we develop and validate a new method of post-processing LRO NAC images of PSRs. We show that our method is able to reveal previously unseen geomorphological features such as boulders and craters down to 3 meters in size, whilst not finding evidence for surface frost or near-surface ice. Our post-processed images significantly facilitate the exploration of PSRs by reducing the uncertainty of target selection and traverse/mission planning.

[1] Max Planck Institute for Solar System Research, Göttingen, Germany. [2] University of Oxford, Oxford, UK. [3] NASA Ames Research Center, Mountain View, CA, USA. ✉email: bickel@mps.mpg.de

Understanding the character and origin of lunar water-ice, and more generally of lunar polar volatiles, is one of seven broad science objectives identified by NASA's Artemis Science Definition Team[1,2] and is of high priority for both science and exploration. Lunar polar cold traps—located within permanently and transiently shadowed regions (PSRs/TSRs)—can trap volatiles delivered from comets, asteroids, solar wind interactions, interior outgassing, and other sources over an extended period of time and are therefore key to understanding the behavior history of volatiles on the Moon as well as other airless bodies in the solar system[3–6]. Evidence for volatiles on or underneath the surface of lunar cold traps, involving varying kinds of modeling and associated assumptions, has been found by the LCROSS impactor mission[7,8], the Lunar Orbiter Laser Altimeter (LOLA[9–11]), the Diviner Lunar Radiometer Experiment (DLRE[12–14]), the Lyman Alpha Mapping Project (LAMP[15]), and the Moon Mineralogy Mapper (M3[16]). However, the scientific instruments used by these studies are limited by their signal-to-noise ratio (SNR) and/or resolution over PSRs, and are only able to provide data and imagery within large PSRs (on a kilometer to sub-kilometer scale). The individual characterization of micro-PSRs, small PSRs/TSRs, and the edges of large PSRs has not yet been possible.

Optical sensors currently provide images of the sunlit lunar surface with the highest spatial resolution, however, the lack of direct illumination makes imaging of PSRs very challenging. Still, PSRs receive small amounts of secondary illumination, i.e., sunlight backscattered from surrounding topography and Earth, as well as faint illumination from stars (e.g., refs. [17,18]), which could potentially be used to image them with optical sensors. NASA's Lunar Reconnaissance Orbiter (LRO) Narrow Angle Camera (NAC) is acquiring regular- and summed-mode optical images of PSRs with spatial resolutions between ~1 and 2 m/pixel, but its short exposure times and low levels of secondary illumination means that CCD and photon noise dominate these images, making any meaningful observations difficult. Over the past decade, LRO's NAC, as well as JAXA's Kaguya Terrain Camera (TC), have used longer CCD exposure times to acquire optical images of PSRs with acceptable SNRs[19–22], however, the spatial resolution of these images (long exposure NAC: ~20–40 m/pixel[20,21] long exposure TC: ~10 m/pixel[22]) is strongly limited by the movement of the satellite and is not able to close the strategic gap posed by the lack of optical, high-resolution images (<10 m/pixel) that are required to study and understand the rover-/astronaut-scale geomorphology and trafficability of PSRs.

The current lack of knowledge of PSRs poses a risk to future ground exploration missions. Small lunar PSRs are top-priority sites targeted by a number of future robotic and crewed missions, for example, by NASA's Artemis program, including the VIPER (Volatiles Investigating Polar Exploration Rover) and Artemis III missions (e.g., ref. [23]). Without high-resolution data and imagery (<5 m/pixel), the meter-scale geomorphology of PSRs remains unknown, and this complicates mission planning and generates questions such as: Where can a given PSR be entered safely? Does it contain hazards to avoid? Where can we take samples or drill? Which PSRs offer the best compromise between risk and science output? Being forced to alter plans during the execution of a mission is inefficient and increases risk, and many problems could be foreseen if the terrain inside the shadows was visible ahead of time.

Here, we address the strategic need for low-noise, high-resolution optical images of PSRs by focusing on improving the quality of the short exposure, full-resolution regular- and summed-mode NAC images. We develop a state-of-the-art low-light denoising tool which we call HORUS—Hyper-effective nOise Removal U-net Software—that is capable of removing the high levels of noise in these images, unlocking the entire LRO NAC image archive for PSR-related science and exploration. The technical details of the algorithm are described in detail in our companion paper[24] and here we focus on demonstrating its value by studying the small-scale geomorphology and trafficability of lunar PSRs for the first time.

## Results

**HORUS—unprecedented views into lunar shadowed regions**. HORUS is a two-stage deep learning-based algorithm that uses environmental information (such as CCD temperature and orbit number) and a physical noise model of the NAC (modified from refs. [25,26]) to remove CCD-related and photon noise from low-light full-resolution LRO NAC regular- and summed-mode images of shadowed regions. We find that HORUS is able to significantly improve the SNR of these images, allowing us to create high-resolution (~1.5 m/pixel) images of lunar PSRs for the first time. HORUS is able to recover features within PSRs as small as 3 to 5 m across; Fig. 1 showcases one HORUS image of a small PSR in the direct vicinity of the lunar south pole. This represents a factor >5–10 spatial resolution improvement over existing long-exposure optical images of lunar PSRs[19–22,27]. Furthermore, as HORUS does not rely on long exposure images that suffer from motion blur and pixel bleed, we are able to produce images of small PSRs < 100 m in diameter (≪0.03 km²), which have not been covered by long-exposure imaging campaigns. In addition to that, HORUS is able to resolve the particularly important sunlight–shadow transition zone (see Fig. 1). Given that small PSRs are significantly more abundant than large PSRs, HORUS represents an unprecedented and rich resource for science and exploration. We refer to the method section, the supplementary material (Fig. S1) and[24] for a detailed description and characterization of the HORUS workflow.

In this study, we focus on demonstrating HORUS' scientific value by denoising and analyzing a series of NAC images of 17 PSRs distributed over five areas of interest (AoIs) across the south pole of the Moon, with shadowed areas ranging from ~0.18 to ~54 km². The selection of AoIs is based on NASA's current array of potential Artemis program landing sites taken from[2]. Specifically, we selected A001 (Shackleton terrace, four PSRs), A004 (Shackleton rim, four PSRs), A011 (de Gerlache terrace, two PSRs), and A102 (Leibnitz plateau, six PSRs)(A = Artemis). We further include one Constellation program AoI, the pyroclastic vent in Schrödinger basin (C′), which hosts a PSR (C = Constellation) (e.g., ref. [28]). The maximum and mean annual surface temperatures across all AoIs range from 96 to 232 K and 64 to 106 K, respectively. Across all AoIs we collect a total of 62 HORUS images, with an average of five and a minimum of three images per PSR. The geographic distribution of all AoIs and PSRs considered is visualized in Fig. 2.

**Validating the unknown**. Examples of noisy raw NAC images of shadowed regions are shown in Fig. 3. Because of the extremely low levels of secondary illumination, CCD and photon noise dominates these images, which manifests itself as stripes and graininess in the image, making any meaningful observations difficult. The images after HORUS denoising are shown alongside the raw images. Whilst HORUS appears to remove this noise effectively, a key challenge for images of PSRs is that there exists no direct ground truth imagery to verify the HORUS-denoised images. To address this we employ multiple strategies to validate the HORUS images and to maximize our confidence when using them for subsequent analysis. These strategies are described below.

Firstly (1), in our companion paper[24], we quantitatively assessed the denoising performance of HORUS on a synthetic

**Fig. 1 HORUS—unprecedented views into lunar shadowed regions.** Example HORUS image of a potential Artemis candidate permanently shadowed region (PSR) (A004 PSR 3) in the direct vicinity of Shackleton crater, lunar south pole. For the first time, we are able to reveal features in this PSR as small as 3–5 m in size. The PSR is outlined in blue, and the NAC-observed (Narrow Angle Camera) transiently shadowed region (TSR) in gray. Intra-TSR/PSR impact craters are clearly visible, just as in the sunlit region; two intra-PSR boulders are highlighted with white circles (both are ~4 m across). Note that the sunlit-shadow transition zone is resolved as well; the range of digital numbers (DN) is shown at the bottom of the figure. The sunlit part of the image is slightly transparent to help showcase the shadowed region. The location of this PSR is shown in Fig. 2. In Egyptian mythology, Horus is the son of Isis; this relation represents the authors' interest in complementing the existing Lunar Reconnaissance Orbiter NAC processing pipeline Isis3 (Integrated Software for Imagers and Spectrometers, e.g., ref. [64]). Raw NAC image credits to NASA/LROC/GSFC/ASU.

test dataset with varying levels of noise. The dataset consists of rescaled sunlit NAC images with synthetic noise added and attempts to closely match the noise conditions expected in shadowed images. Noise is generated by combining real noise samples from dark NAC frames with a physical noise model of the NAC CCD. In[24], we then compared the HORUS denoising performance on this dataset to the default Isis3 lronaccal routine and a standard frequency domain-based filter, using L1 error, PSNR (peak signal-to-noise ratio), and SSIM (structural similarity index measure) as metrics. We observed that HORUS significantly outperforms all other routines on all metrics across all noise levels, reliably improving SNR. A key observation from this study is that the denoising performance of HORUS strongly depends on the level of secondary illumination, i.e., the number of photons received. In particular, the minimum feature size HORUS can recover strongly depends on the photon count (Fig. S2). For the synthetic dataset and photon counts around 10–15 digital numbers (DN), the minimum feature size resolvable is ~3 to 5 m. In[24], we observed that this feature size increases significantly for lower counts (i.e., only larger features are resolvable) and decreases slowly for higher counts (i.e., smaller features become resolvable). We observed that HORUS tends to remove features below the minimum feature size (i.e., introduces false negatives), but does not introduce any false features (i.e., does not introduce false positives).

Secondly (2), we qualitatively and quantitatively assess HORUS' performance using TSRs. In these regions, we take sunlit images as ground truth and compare them to HORUS denoised shadowed images. An example comparison is shown in Fig. 3. This comparison indicates that HORUS is able to preserve physical features, such as boulders and craters, in these images as small as ~3 m in diameter. Furthermore, similar to the synthetic

observations above, HORUS does not introduce false features (i.e., does not introduce false positives). We are also able to use TSRs to carry out application-specific validation, for example when carrying out crater counting within a PSR we also carry out crater counting within a neighboring TSR as a control test. Our application-specific TSR validation methods are described in more detail below.

Thirdly (3), when analyzing each PSR, we qualitatively assess performance by comparing multiple HORUS frames over the PSR. Here we assume features which appear in more than 2 HORUS frames, i.e., features which are spatially and morphologically consistent, are in fact physically present (Fig. 4). We sometimes observe that faint residual noise is left by HORUS (as seen in Figs. 3 and S4 through S20), and the cross-comparison of HORUS PSR frames allows us to discard this noise which could otherwise have been misinterpreted as surface albedo variations or very small (<~3 m in diameter) physical features.

Fourthly (4), we generate SNR maps for specific PSRs by dividing the mean photon signal (in DN) estimated by HORUS by the standard deviation of the noise removed by HORUS, averaged over a 100 m radius. These maps are useful for quantifying our confidence in the HORUS denoising (Fig. S3) alongside the validation approaches above, as they give us a direct measure of how difficult the image is to denoise. Similar to the DN counts, lower SNR values indicate that the minimum size of resolved features increases. The smallest observed features (3 to 5 m in diameter) are typically observed in HORUS images with SNRs larger than five.

Validation steps (2) and (3) have been applied for all images used in this paper. More detailed discussions on the validation results specific to the scientific findings of this work are presented below. All of the validated HORUS images of PSRs used in this study are displayed in the supplementary information (Fig. S4 through S20).

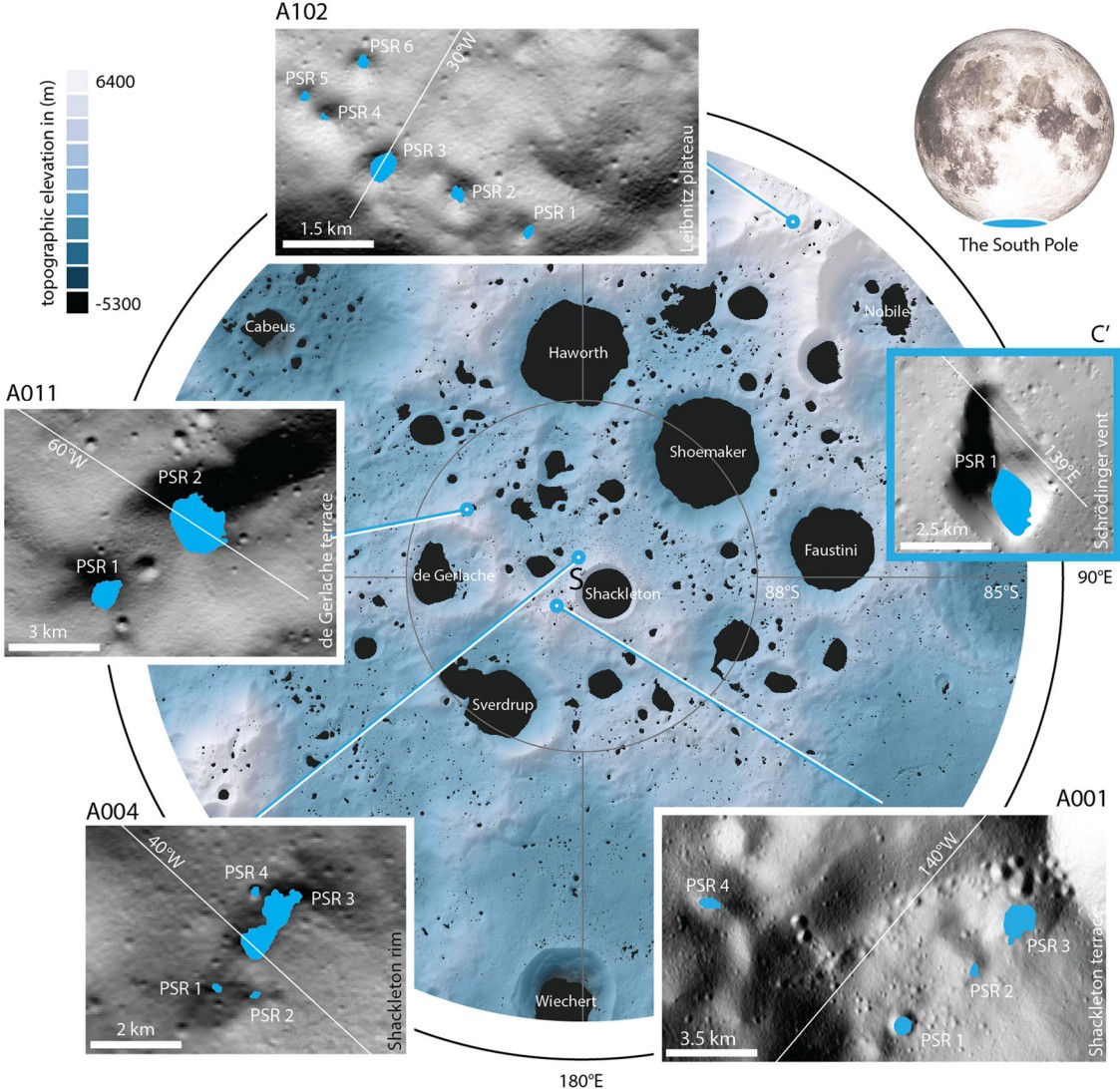

**Fig. 2 Overview of the study areas.** Topographic map of the lunar south pole, including large and small PSRs (permanently shadowed regions, black polygons); insets show the five sites studied in this work (LOLA hillshade in the background). The insets use blue polygons to mark PSRs. All sites are either potential Artemis program candidates (A[2]) or were Constellation program candidates (C′). The site in Schrödinger Basin is distal and not indicated on the map (indicated by the blue box). PSR locations and extent taken from ref. [65].

**The boulder population of small lunar shadowed regions.** We first use the HORUS images to map boulders across all the target AoIs and PSRs. Examples of identified boulders are shown in Figs. 1, 3, and 4. Across the PSRs we identify a total of 20 intra-PSR boulders with average, minimum, and maximum diameters of 10, 3, and 19 m respectively. Not all PSRs host boulders and the spatial density of boulders is generally very low, around ~4 boulders/km², on average; we only find boulders in C′, A004 PSR 1, and A004 PSR 3. Boulders are located close to the edge of PSRs as well as towards their centers.

We perform a validation test within a TSR that is located right next to A004 PSR 3, mapping boulders (n = 5) in sunlit (regular NAC image) and shadowed conditions (HORUS image). The used HORUS image is the same as used for the intra-PSR boulder count of A004 PSR 3. The small number of boulders is governed by the overall scarcity of boulders in the studied TSRs. We observe that boulders that are observable in sunlit and shadowed conditions appear slightly larger in the considered HORUS image (shadowed conditions), roughly ~0.9 m larger on average (~11% larger relative to boulders observed in sunlit conditions), which is less than one pixel, meaning that HORUS might potentially introduce a subtle observational bias, at least for small, high intensity features such as boulders, depending on the DN and SNR of the respective HORUS image.

It is unclear how these boulders have been deposited in their host PSRs, i.e., if they have a local or distal (e.g., rockfall/impact) origin. Despite the size of the observed boulders, we do not observe any boulder tracks and/or imprints to prove their distal origin. The lack of tracks could indicate that boulders never moved or were deposited more than ~1.5 to 35 Ma ago, i.e., that tracks have been eroded through processes such as micrometeoroid bombardment (e.g., refs. [29,30]). In contrast, we do not observe any signs of boulder degradation, such as fragments or fillets, suggesting that the observed boulders are not particularly old. We note that the surrounding terrain of the respective PSRs generally favors the occurrence of mass wasting, with slope angles ranging from ~20 to more than ~30° (see Fig. 2 and S21). Previous studies have reported rockfalls, recent impact events, and potential boulder source regions in the direct vicinity of these sites that could have produced and/or transported boulders[31–33]. Assuming the observed intra-PSR boulders have a distal origin, their presence could be an indication for the mechanical

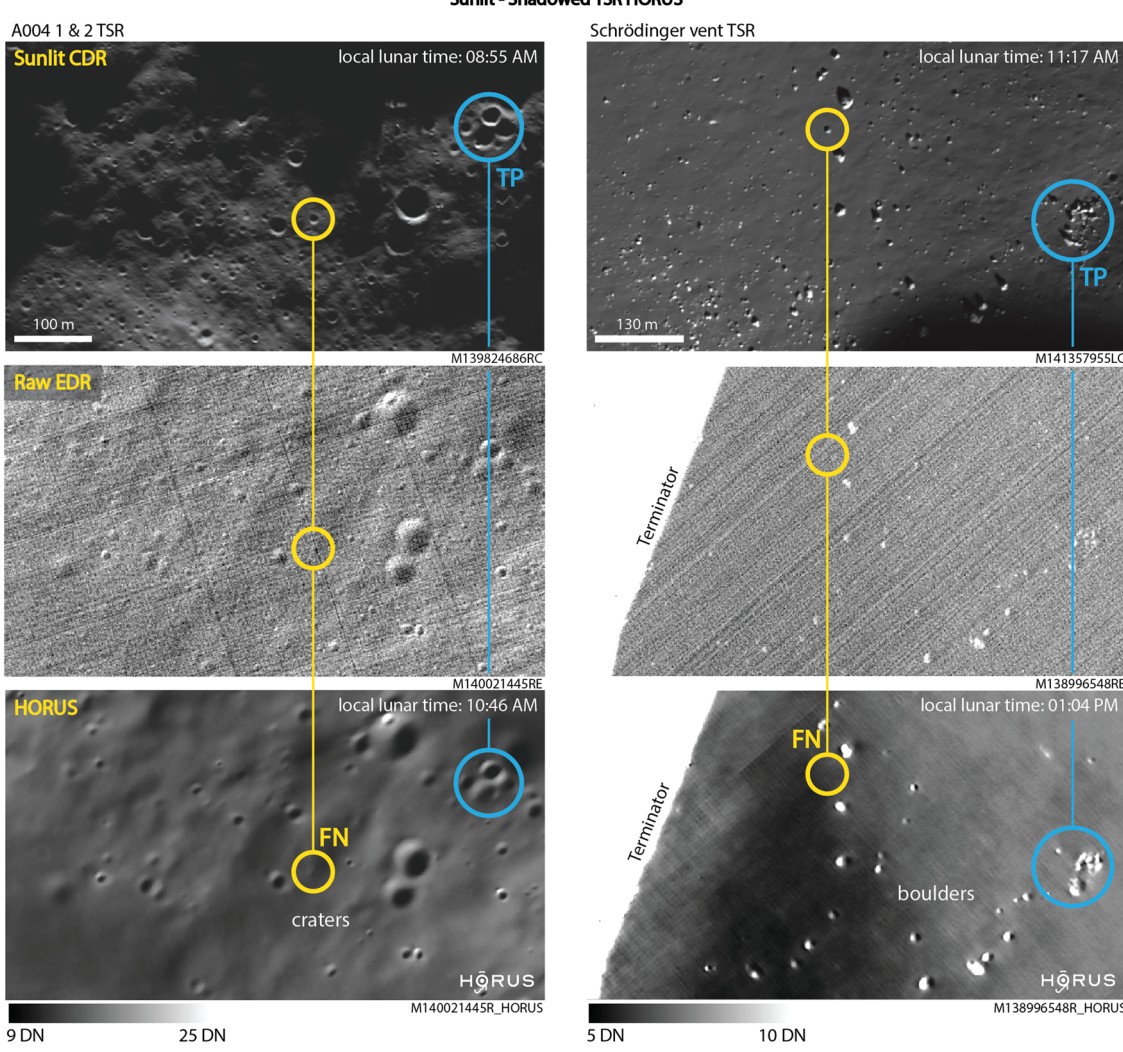

**Fig. 3 Transiently shadowed regions validation of HORUS images.** Example HORUS images in two different TSRs (transiently shadowed regions); we compare HORUS denoised shadowed images to corresponding shadowed raw NAC EDR (Narrow Angle Camera Experimental Data Record) and sunlit CDR (Calibrated Data Record) images of the same region. We observe that HORUS can image features visible in the corresponding sunlit image down to a certain minimum feature size (TP, true positive, blue). Features that are smaller than this appear faint or do not appear in HORUS at all (FN, false negative, yellow). This minimum size depends strongly on the photon count (digital number, DN). In the examples above, the smallest resolved crater diameter is ~5 m and the smallest resolved boulder diameter is ~4 m. We do not observe false positives other than residual noise. Raw NAC image credits to NASA/LROC/GSFC/ASU.

properties of the shallow subsurface of the respective PSRs. However, it is important to note that boulders with diameters greater than 3 m usually carve tracks more than ~0.75 m deep[33,34], meaning that the mapped boulder population (with a mean boulder diameter of ~10 m) would not allow for the study of the mechanical properties of the near surface of PSRs, but of their subsurface, between an estimated ~1.5 and ~2.5 m depth.

While mapping boulders in the PSR of Schrödinger basin's pyroclastic vent (C′), we discovered two clusters of boulder-like, high-reflectance features high up on the northern wall of the vent. Those features are located on the exact same topographic elevation as the high-albedo rock outcrops on the vent's sunlit eastern and western wall, and thus could be outcrops of the same lithological unit. We note that these potential outcrops are candidate source regions for some of the boulders and scree within the PSR. An overview map of the mapped features in the vent is presented in Fig. S21.

**Crater counting and cumulative crater size-frequency distributions.** Next, we map the locations and diameters of all observable intra-PSR craters across all AOIs. The diameters of craters range from ~4 to ~320 m, with an average of ~18 m. The spatial density of craters across our AoI PSRs ranges from ~0.1 to ~68 craters/km², with an average of ~15 craters/km². The observable craters in the studied PSRs cover between ~0.03 and ~12% of their host PSRs surface, with a mean of ~4.6%. We recognize that varying levels of secondary illumination across the different PSRs (and hence the minimum feature size resolvable by HORUS) could potentially influence the observed crater density variations. The majority of the observable craters in our AoI PSRs do not appear to overlap, potentially allowing for a traverse around them. We further note qualitative differences in crater appearance, regarding feature albedo, sharpness, and texture, which might represent differences in crater formation age. Figure 5 showcases one such potentially fresh impact crater; Table S1 lists all derived numerical values per AoI and PSR.

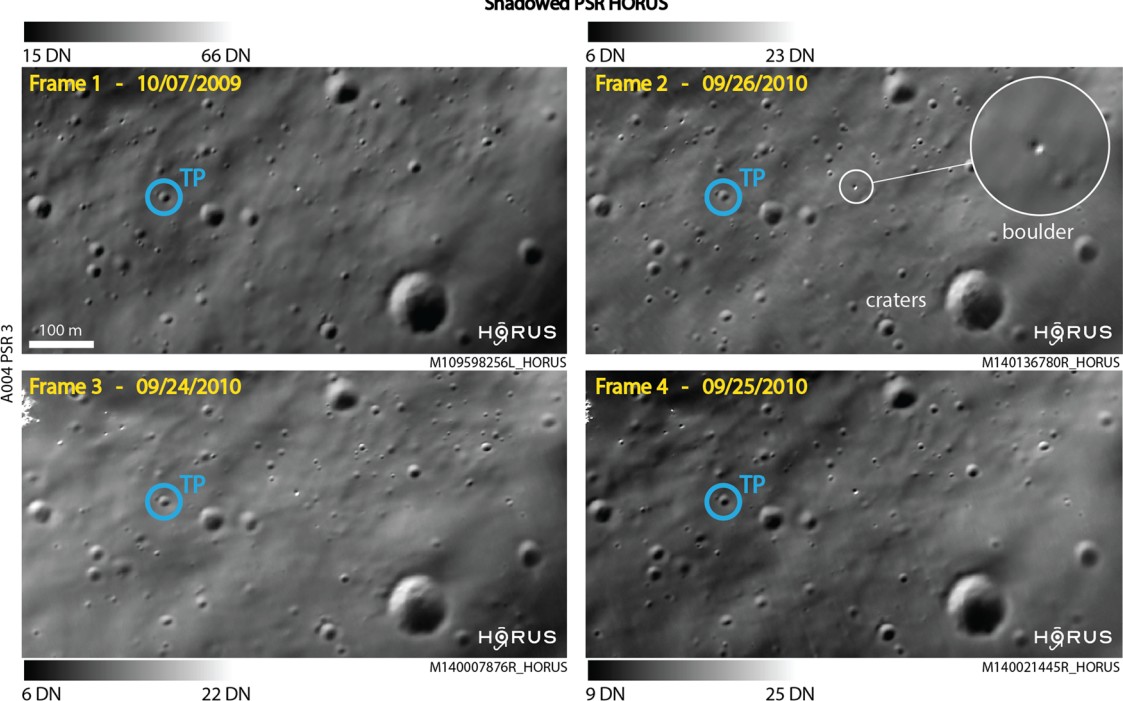

**Fig. 4 Permanently shadowed regions validation of HORUS images.** We compare multiple HORUS frames over the same PSR (permanently shadowed region). The vast majority of features appear consistent across all HORUS frames, i.e., they are shape-, dimension-, and location-consistent, and are thus considered to be physical (TP, true positives, blue). For this study, features are only considered physical if they appear in at least two HORUS images. In the examples above, the smallest resolved boulder diameter is ~4 m; the range of digital numbers (DN) is shown next to each panel. This figure also illustrates how HORUS can be used to study the temporal variability of PSRs on timescales of days and years, within the limitations of the method. Raw NAC (Narrow Angle Camera) image credits to NASA/LROC/GSFC/ASU.

Next, we derive the crater size-frequency distributions (CSFDs) of five of our larger and/or well-secondary-lit PSR AoIs (A001 PSR 10, A004 PSR 1, A004 PSR 2, A004 PSR 3, and A102 PSR 6) and compare them to the CSFDs of five equally sized, sunlit regions in the direct vicinity of their respective PSR. Despite the fact that we are able to recognize very small craters (down to ~4 m in diameter) in the selected PSRs, the derived CSFD curves indicate that the investigated PSRs generally feature fewer craters with diameters smaller than ~8.5 to 11 m than their immediate sunlit surroundings (Fig. S22).

This could indicate that either: (A) there is a physical process—absent in sunlit regions—that is erasing small craters in lunar PSRs, for example, volatile/dust migration or volatile sequestration as discussed in the literature (e.g., refs. [35–39]). (B) the upper layers of regolith in TSRs/PSRs feature anomalous geomechanical behavior e.g., caused by an increased porosity—as predicted by a number of remote sensing studies (e.g., refs. [15,40–42]) as well as the LCROSS impactor[8]—leading to increased impact crater diameters, as has been reported for e.g., pyroclastic deposits[43]. Alternatively, (C) there is an observational bias in the used HORUS images, i.e., even if very small craters can be recognized locally, not all physically present small craters across the PSR are visible, e.g., due to spatial and subtle variations of the secondary illumination and SNR. We perform an additional validation test within a TSR that is located within A004, in between PSRs 1, 2, and 3 (89.82°S and 44.01°W). We map all craters in a predefined area in sunlit (regular NAC CDR image) and shadowed conditions (the same HORUS image used for the intra-PSR CSFD curve) and compute various CSFDs (shown in Fig. 6). We draw two conclusions from this test: the first is that for craters which are observable in both images, the crater diameter measurements performed in HORUS images do not suffer from

any observation bias; in other words, HORUS does not appear to increase the diameter of craters compared to the sunlit image (Fig. 6 left). The second is that HORUS does not reliably resolve a substantial fraction of the crater population with diameters smaller than ~8.5 m; in other words, HORUS contains some, but not all craters < ~8.5 m in diameter (Fig. 6 right); we note that the observed threshold diameter could be subject to change, depending on the DN and SNR of the used images. This leads us to discard hypotheses (A) and (B), although we acknowledge the large statistical uncertainties associated with crater counting in such small AoIs, which is a physical limitation of the study of small PSRs, however. The analysis of additional, small PSRs will help to gather more confident statistics.

We observe that the AoI PSR C′ in Schrödinger's pyroclastic vent features an anomalously low spatial concentration of craters (~0.07 crater/km$^2$, compared to the overall AoI mean of ~15 craters/km$^2$), meaning that only ~0.03% of the PSR area are cratered (in contrast to the overall AoI mean of ~4.6%). The analysis of the DN count and SNR distributions (DN ranges between ~5.8 and ~12.1, while SNR ranges between ~1.4 and ~2.4) of the used HORUS images indicates that (1) a spatial bias as discussed above is likely and (2) there is a substantial chance that not all physically present features are resolved. However, it is unclear if the limited SNR alone is responsible for the nearly complete lack of craters, especially as multiple boulders and a few craters can be observed across the PSR (see Figs. S20 and S21).

**HORUS-enabled traverse planning.** Besides studies of geomorphology, HORUS can also be used to inform traverse planning. We enhanced our HORUS images by adding topographical information from LOLA and used both datasets to lay out a hypothetical rover traverse that targets AoI A102 PSR 6, a ~450 m

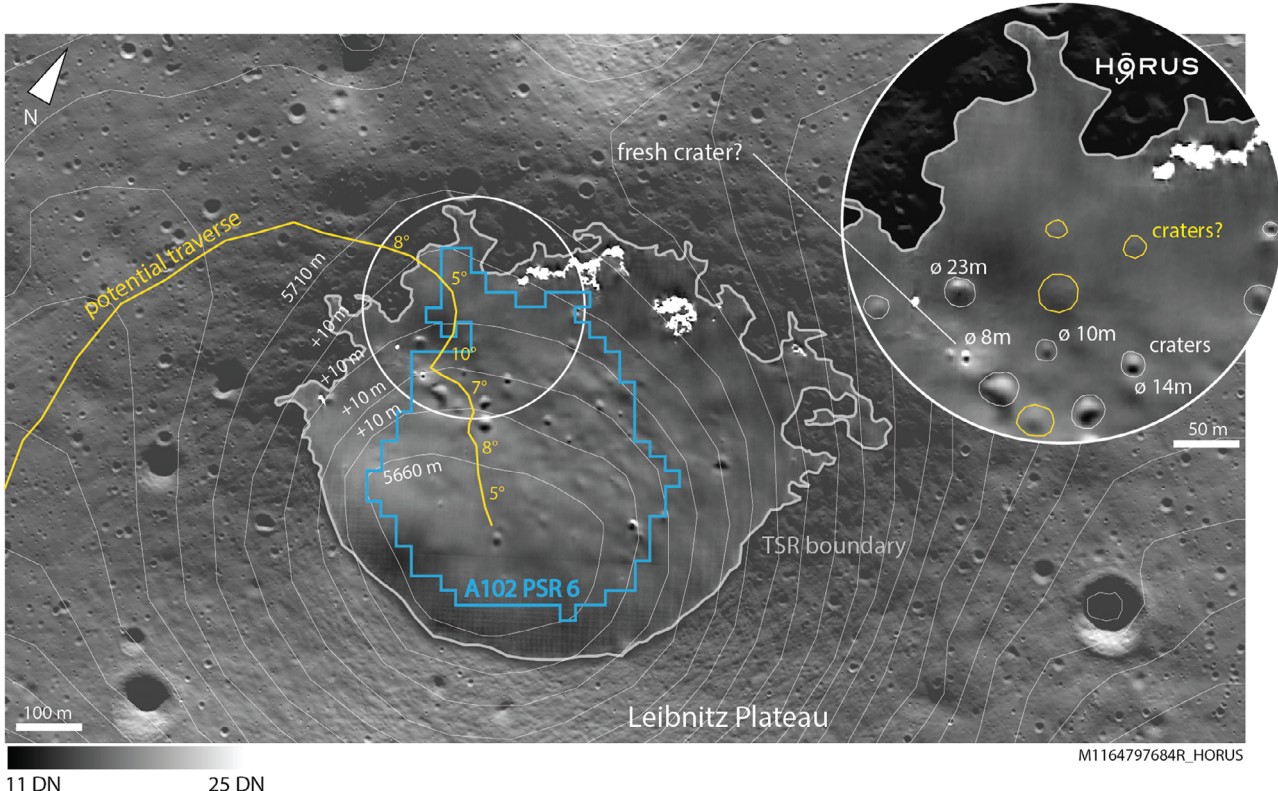

**Fig. 5 Using HORUS to plan a traverse into a small permanently shadowed region.** HORUS image of a potential Artemis candidate PSR (permanently shadowed region) on the Leibnitz plateau, embedded into a regular NAC (Narrow Angle Camera) image. Topography is indicated with white contour lines; each line corresponds to a 10 m elevation change. The extent of the PSR is indicated in blue, using a 20 m PSR product calculated for this particular PSR, and the boundary of the NAC-observed TSR (transiently shadowed region) is indicated in gray. The inset shows a potential access point in more detail, including a potentially fresh (bright) impact. A potential access traverse is shown in yellow, including local slope angles which never exceed ~10°. The sunlit part of the image is slightly transparent to help showcase the shadowed region; the range of digital numbers (DN) is shown at the bottom of the figure. Raw NAC image credits to NASA/LROC/GSFC/ASU.

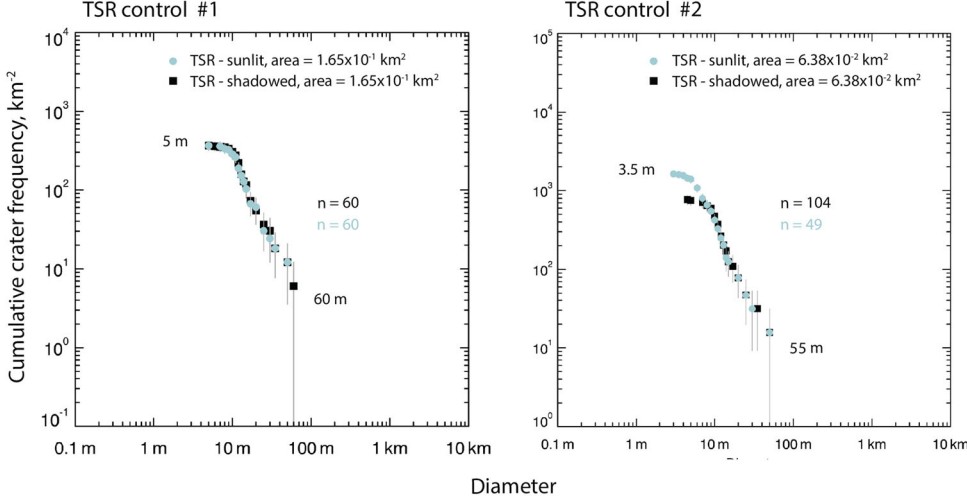

**Fig. 6 Crater size-frequency distributions (CSFD) for a transiently shadowed region (TSR) control site (Latitude 89.82°S Longitude 44.01°W) next to A004 PSR 3.** Left: Controlled comparison of two cumulative CSFD curves with 60 craters (log-log plot); controlled means that only craters are considered that are visible in sunlit (blue curve, M140007876RC) and shadowed conditions (black curve, M109598256L_HORUS). Both curves match well, indicating that crater diameter measurements performed in HORUS images appear to not introduce a significant observational bias. Right: Uncontrolled comparison of two cumulative CSFD curves (log-log plot); uncontrolled means that all visible craters within the area and images are considered (sunlit: blue curve, M140007876RC; shadowed: black curve, M109598256L_HORUS). The curves indicate that HORUS introduces an observational bias for crater diameters < ~8.5 m, i.e., some craters with diameters < ~8.5 m can be retrieved, but not all of them. Image M109598256L_HORUS is the same image used for the intra-PSR (permanently shadowed region) CSFD curve as discussed in the text (see Fig. S22). The error bars correspond to the statistical error of the cumulative numbers ($\sqrt{N}$).

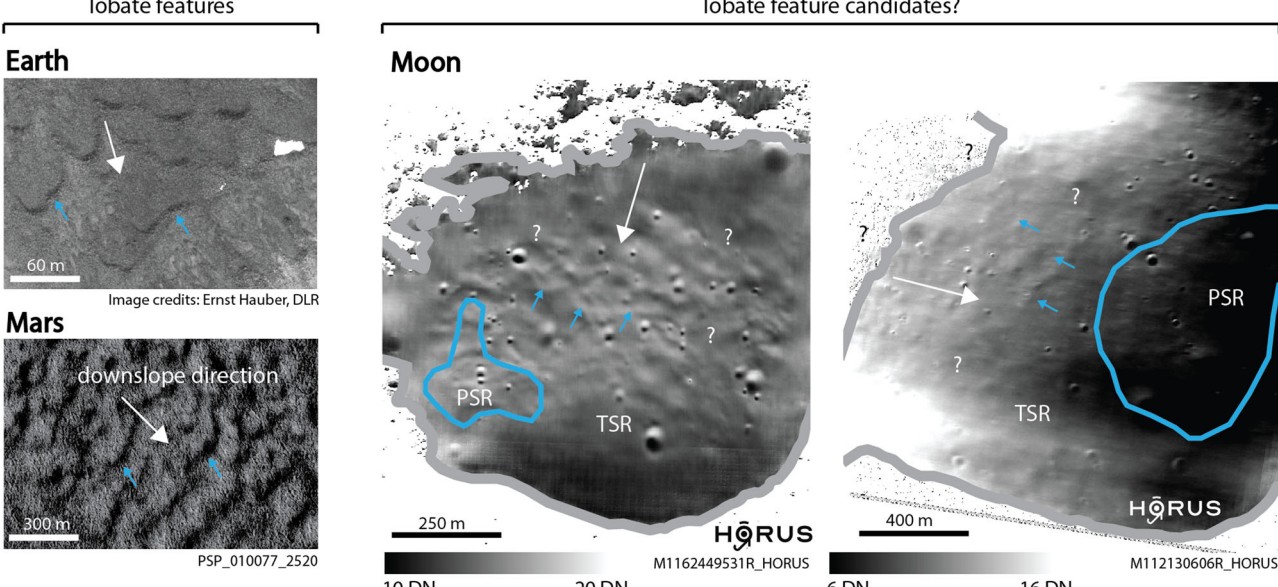

**Fig. 7 Ice-related, geomorphic features on Earth and Mars—and the Moon?** Examples of geomorphic expressions of ice-bearing soils, such as lobate features (solifluction lobes), found on Earth (here: High-Resolution Stereo Camera AX (HRSC-AX) image, Svalbard, Arctic Norway[50]) and potentially on Mars (High-Resolution Imaging Science Experiment (HiRISE) image[51]). Lunar TSRs (transiently shadowed regions) might potentially host similar geomorphic features, as indicated on the right-hand side (the boundary of the NAC-observed (Narrow Angle Camera) TSR is indicated in gray; the PSR (permanently shadowed region) boundary is indicated in blue, selected lobate-type features are indicated by blue arrows, for A102 PSR 5-left and A011 PSR 1-right). The local slope is indicated with a white arrow. HORUS images have been contrast stretched to highlight the candidate features; the range of digital numbers (DN) is shown at the bottom of the figure. HRSC-AX image credits to Ernst Hauber (DLR); HiRISE image credits to NASA/JPL/UoA; raw NAC image credits to NASA/LROC/GSFC/ASU.

diameter PSR on the Leibnitz plateau, close to the rim of the Nobile crater (Fig. 5). The analysis of HORUS images revealed a high-albedo and potentially fresh impact crater within this PSR, which could represent a promising science target. HORUS images and slope information indicate that an access from the west might be safest in terms of topography and obstacles: the traverse would not feature slope angles steeper than ~10°, and the traverse could be planned in a way that helps the rover to avoid all visible physical obstacles. After examination of the potentially fresh crater and its ejecta blanket, the notional traverse continues to lead the rover deeper into the core of the PSR, following the gentle, local topographic gradient while avoiding a series of impact craters scattered across the PSR. We note that HORUS products could be applied—in a similar way as presented here—to inform and guide traverse planning in any other lunar PSR.

**Potential indications for (near-) subsurface ice abundance.** Finally, we closely examined HORUS images of all AoI PSRs for evidence of surface frost or ice, such as e.g., bright patches or lenses, although we could not find any such evidence. The absence of surface frost might be expected as the majority of AoI PSRs feature maximum annual surface temperatures slightly above 100 to 112 K where water-ice would be stable for an extended period of time[13,44–47], but evidence for surface frost or ice is also absent in the two PSRs with lower temperatures (A001 PSR 3 and A011 PSR 2). Impact craters across our AoIs do not appear to expose any subsurface ice, although any exposed ice might have sublimated before it was observed by LRO. Our observations are in agreement with earlier studies of long-exposure optical images of PSRs that did not find any evidence of large-scale surface frost either[20,22,27,48]. The apparent absence of surface frost and ice patches could indicate that ice may only exist in the shallow subsurface of the studied PSRs, may be intermixed with regolith as fines, or might be present as a (dis)continuous

matrix (e.g., refs. [37,49]), which would explain why some (non-optical) instruments register the footprint of water-ice on the surfaces of PSRs[13,16] despite the lack of evidence in optical images. Due to that mixing and the limited spatial resolution of NAC/HORUS, the scarce secondary illumination might not be sufficient to make ice/frost-caused albedo differences visually apparent in optical images, as observed in the case of long-exposure images (e.g., ref. [19]).

Even if visually absent, the general presence of ice on or underneath the surface might be revealed by larger-scale, geomorphic features that are characteristic for ice-bearing soils, such as ice wedges, patterned ground, and lobate features, as observed on Earth (e.g., ref. [50]) and Mars (e.g., ref. [51]), for example. We observe slope-orthogonal lobate features, which bear visual resemblance to terrestrial and martian solifluction features—sometimes referred to as elephant-hide terrain, creep, or textured ground[52–55], that extend from predominantly sunlit regions into TSRs (predominantly shadowed) (Fig. 7), but appear to be absent in PSRs. We do not observe obvious changes in the morphology of these features, such as width and spacing, as a function of their location within a TSR or their proximity to a PSR. Within TSRs, lobate features appear to be exclusively located on slope angles ranging from ~10 to ~20° (as observed in AoIs A001 PSR 1, A001 PSR 2, A004 PSR 3, A011 PSR 1, A011 PSR 2, A102 PSR 3, A102 PSR 4, A102 PSR 5, and A102 PSR 6).

The transiently/seasonal illumination of potentially ice-bearing regions could lead to sublimation of (sub)surface ice (see e.g., ref. [38]). Vertically constrained by a potential underlying, impermeable ice layer or ice-cemented breccia or regolith, the sublimation process could lead to a local, short-lived modification of the geomechanical properties of the regolith, which could ultimately result in a lateral displacement of material, potentially facilitated by local slopes. Assuming that the presence and sublimation of (sub)surface ice would affect the mechanical

properties of the regolith, the apparent lack of morphological differences between sunlit- and TSR lobate features could indicate that (1) the predominantly sunlit polar regolith contains a similar amount of (shallow) subsurface ice as the predominantly shadowed polar regolith (TSRs); or (2) both regions are similarly depleted of shallow (sub)surface ice. Further, the apparent lack of lobate features within PSRs might on the one hand indicate that (3) lobate features indeed are only related to gravitational creep without the involvement of ice, given that small PSRs are usually located in relatively flat, topographic depressions (resulting in a lack of slopes); or on the other hand that (4) the lack of seasonal illumination in PSRs is the reason for the absence of sublimation and thus surface displacement/ lobate feature formation.

We do not expect HORUS to remove or hallucinate lobate-type features, as elongated lobes and ridges are common geomorphic features across the Moon and, thus, should be well represented in HORUS' training dataset. We perform a qualitative validation test within a TSR that is located close to A001 PSR 10 (89.59°S, 118.73°W), comparing the visual appearance of surface texture and lobate-type features in sunlit (regular NAC image) and shadowed conditions (HORUS image). We observe that—overall—the analyzed HORUS images appropriately display the surface texture and relevant features as observed in sunlit conditions (Fig. S23). However, we note that varying illumination conditions might highlight different aspects of the surface texture present in a PSR, as is the case for sunlit regions as well. In this study, we also use multi-frame image validation, which allows further verification by requiring features are consistent across the frames, which typically have varying illumination conditions (see Figs. 3 and 4). The fact that similar lobate-type features have not been observed in low-resolution, long-exposure images of TSRs and PSRs might be explained by their small size—we note that there may be other ice-related geomorphic features smaller than the spatial resolution as currently offered by HORUS. We note that additional, systematic observations are required to better understand the potential role of ice and ice sublimation in the formation of lobate-type features and other small-scale geomorphology, in shadow as well as in sunlit polar regions. We point out that, if the potential role of ice in landform generation can be confirmed, the presence of ice-related landforms, as well as the potentially sublimation-modified geomechanical properties of the affected regions, might make access to PSRs more challenging. In turn, the presence of such features would imply that in those regions ice is—or has been—relatively close to the surface and might be easier to access by exploration and ISRU missions.

## Discussion

By denoising regular- and summed-mode LRO NAC images, HORUS provides low-noise, high-resolution images of lunar shadowed regions, closing an important strategic knowledge gap. Importantly, HORUS allows us to tap into the large and growing archive of more than 200,000 NAC images over the lunar north and south pole (beyond 80°N and S). We observe that roughly 25% of the NAC images covering the AoIs considered for this work had useable signal allowing HORUS post-processing to be used, meaning that there could be an estimated total of ~50,000 HORUS images available, enabling studies of an extensive number of small and large shadowed regions, including studies of the temporal and seasonal variations of TSRs and PSRs at an unprecedented temporal and spatial resolution, i.e., on timescales from days to years (Fig. 4). Applying HORUS to a raw NAC image takes less than a minute if graphical processing units (GPUs) are used, meaning that HORUS could be deployed at scale to produce a large archive of denoised images of lunar shadowed regions. HORUS allows us to test hypotheses about the potential connection between PSR surface brightness variations, hypothesized/modeled ice content, large- and small-scale topographic roughness, and surface-subsurface ice abundance, as discussed by[3,27,36,56,57], and[58], among others. Furthermore, HORUS is not only useful for studying true PSRs, but can also enable the analysis of TSRs which were never photographed in sunlight, due to orbital or other constraints. Whilst effective, it is important to remember that HORUS still needs some photon signal to be present, and given that polar topography governs the amount of secondary illumination in PSRs, some PSRs may be more easily studied with HORUS than others. Furthermore, our validation suggests that whilst HORUS is able to resolve small features in PSRs, depending on the SNR available it may not be able to fully resolve all small features present.

The findings of this study have practical implications for the imminent exploration of PSRs. The lack of direct, visual evidence of frost or ice on the surfaces of small PSRs seems to confirm earlier findings that volatiles—if present—are dispersed and that large concentrations of ice and other volatiles might only be accessible underneath the surface, i.e., require trenching, drilling, or other types of interaction with the surface. Work-based on Mini-RF Radar data[59,60] suggested that the presence of surface frost and buried ice deposits in PSRs could be mimicked by surface roughness and rocks (scree), however, our observations made from HORUS images in small PSRs indicate that the respective rocks—if present— would need to be completely buried or much smaller than ~3 m in diameter, at least in the studied small lunar PSRs.

Our preliminary observations do not provide geomorphic evidence for a shallow high porosity surface layer, as could be revealed e.g., by systematically increased crater diameters, but offer potential additional evidence for substantial bearing capacity at depths greater than ~0.5 m, as suggested for low-latitude PSRs by[48]. It remains unclear to what extent the geomechanical properties of the shallow and deep umbral regolith are controlled by volatiles and the environmental conditions in PSRs. There are regolith-ice configurations which could provide sufficient bearing capacity and traction even in highly porous regolith, such as ice-cemented regolith, ice breccias, and ice-matrix breccias[37,48,61] that might be benign to exploration. Figure 8 features a synopsis of information derived by this study as well as by earlier studies. We point out that the number of the sites studied in this work is limited and that AoIs are scattered across the pole. In addition, it is important to note that information about the boulder and crater population < ~3 m is missing and might change the findings of this study.

Despite our extensive efforts to validate HORUS images, many of the initial findings presented here depend on verification and ground truth. Future missions such as NASA's VIPER mission, the ShadowCam instrument on Korea's Pathfinder Lunar Orbiter, and the Lunar Trailblazer mission, to be launched in late 2023 and in mid-2022, respectively, will be able to provide such ground truth[23,62]. In turn, HORUS could augment data products from future missions, such as ShadowCam, where data could be incorporated in the training dataset of HORUS to improve its overall performance, or HORUS could be adapted to enhance ShadowCam images themselves, potentially producing a series of low-noise, high(er)-resolution images of lunar PSRs that spans over a decade—and potentially more.

## Methods

**HORUS setup, training, testing, and validation**. The HORUS denoising workflow is shown in Fig. S1. HORUS consists of two deep neural networks which are applied in sequence to remove different noise sources in the summed- and regular-mode LRO NAC images. The input to HORUS is a raw NAC EDR (experimental data record) image along with a set of metadata recorded at the time of image

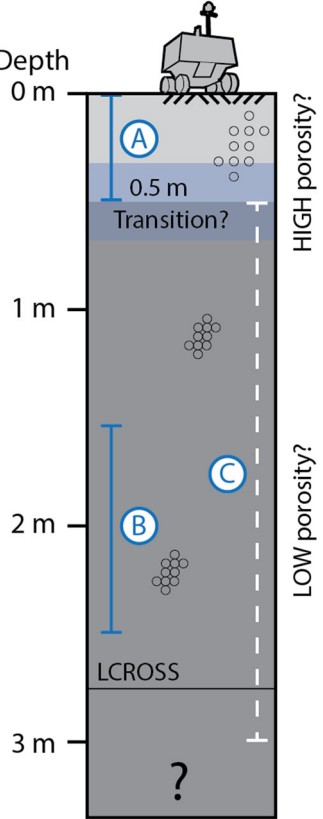

**Fig. 8 Permanently shadowed region regolith surface and subsurface characteristics.** Sketch of inferred lunar PSR (permanently shadowed region) surface and subsurface characteristics, based on this work and literature (see discussion). **A** Layer of potentially highly porous regolith, as suggested e.g., by FUV-dark (far ultra-violet) PSR surfaces and the lack of illumination-driven densification processes in PSRs[8, 15, 40–42], < ~0.5 m depth; **B** layer of high shear strength regolith as indicated by potentially distal, rockfall/impact-deposited intra-PSR boulders (> ~1.5 to 2.5 m depth); **C** depth range where high intra-PSR shear strength regolith has been reported in low-latitude PSRs, but with some uncertainty about the depth estimates caused by secondary illumination[48]. The LCROSS (Lunar CRater Observation and Sensing Satellite) Centaur stage created an impact crater with an estimated diameter of 25 to 30 m[8]; using a d/D ratio of 0.1 as suggested by[69], LCROSS would have reached a depth of ~2.5 to 3 m, i.e., represents the deepest sample of a PSR to date. Rover not to scale.

capture and the final output of HORUS is an estimate of the mean photon signal contained within the image. The first network, called DestripeNet, takes the metadata as input (which consists of the values of the NAC CCD masked pixels, and the other metadata listed in Fig. S1), and outputs a prediction of the CCD dark noise components (such as the dark bias and dark current noise) contained in the image. The network uses a convolutional decoder architecture and is trained in a supervised fashion using over 70,000 dark calibration frames as labels, which were captured over the entire duration of the LRO mission. The predicted dark frame is subtracted from the image, and the nonlinearity and flatfield corrections from ref. [25] (as used in the current Isis3 calibration routine) are then applied. This intermediate image is input to PhotonNet, which estimates the residual noise sources in the image, namely the CCD read noise, compression noise (generated by the compression and subsequent decompression of the EDR image before down-link to Earth), photon (shot) noise, and any residual dark noise which DestripeNet failed to predict. This residual noise estimate is subtracted from the intermediate image, resulting in a clean image which estimates the mean photon signal in DN. PhotonNet uses a U-Net[63] architecture and is trained using a synthetically gen-erated dataset of noisy-clean image pairs, where clean images are generated by rescaling many randomly selected NAC images of sunlit regions to low photon counts, and noisy images are generated by adding noise from a physically realistic noise model of the NAC CCD. To ensure the training distribution of PhotonNet is as close to the test distribution of PSR images as possible, 3D ray tracing was used to select clean training images that had similar solar incidence angles to the angle of incidences expected from secondary illumination in PSRs. We refer the interested

reader to[24] for a more detailed description and characterization of the HORUS denoising workflow.

We validated HORUS using the four approaches discussed in the main body of this paper: (1) a quantitative performance on synthetic data, (2) a cross-comparison of sunlit and shadowed images of TSRs, (3), a cross-comparison of multiple shadowed images of PSRs, and (4) SNR mapping. Validation approach (1) was performed for the HORUS workflow in general, while approaches (2) and (3) have been performed for all images used in this study. Approach (4) has been used for images that cover sites of increased interest, such as sites where intra-PSR crater counting has been performed. We note that approach (4) is not a validation approach by itself, but is useful for increasing our confidence in the images in combination with the other approaches. The SNR maps are derived by determining:

$$\mathrm{SNR} = \frac{\mathrm{mean}(N(S))}{\mathrm{std}(I - N(S))}, \tag{1}$$

Calculated over a sliding 2D Gaussian-weighted window with a standard deviation (width) of 100 m, where $S$ is the mean photon signal image output by HORUS (in units of DN), $I$ is the raw EDR image (in units of DN), and $N(\bullet)$ is the forward nonlinearity function used in ref. [24].

**Image processing and projection.** We used compute resources on the Google Cloud Platform, specifically a virtual machine with 24 CPU cores, 200 GB RAM, and 8 NVIDIA GPUs (Tesla V100). The image data was downloaded from the Planetary Data System (http://lroc.sese.asu.edu/data/). Training both HORUS networks took ~12 h, and the processing time for running HORUS after training on a single NAC EDR 52224 × 5064 image was about 30 s using one NVIDIA Tesla V100 GPU. After applying HORUS, we output the images as .cub files and used Isis3 (e.g., ref. [64]) (cam2map) to map-project all images to a polar stereographic projection (.geotiff). We note that downloading the raw image as well as the map-projection of the denoised images are significantly more time-consuming than the processing of the images in HORUS itself. We are currently able to output ~30 full map-projected HORUS images (.geotiff) per hour using four GPUs.

**Crater counting and CSFD curve comparison.** Crater counting was performed on validated HORUS images in QGIS (https://www.qgis.org/de/site/) using the CADDigitize tool by L. Bartoletti, specifically using the two-click circle tool. Here, a first click defines the center of a crater and a second click defines its rim. We consider the accuracy and precision of our mapping to be less than one pixel, i.e., < ~1.5 m. Care has been taken to not include secondary craters as well as unrelated, potentially round geomorphological features. Where possible, we selected the most appropriate images to map craters, i.e., images with the most appropriate illumi-nation conditions to recognize negative relief features.

Our AoI PSRs have areas ranging from 0.18 to 54 km²; all sunlit control areas have the same extent as their associated intra-PSR area and are located in their direct vicinity, governed by the illumination and the available NAC images. The location and extent of all PSRs is taken from ref. [65], who performed illumination modeling based on LOLA topography and provide polar PSR maps with spatial resolutions of 60 and 120 m/pixel. We require a PSR to cover at least 3 by 3 pixels in the 60 m/pixel map presented by ref. [65] to make sure we are only studying true PSRs. Care has been taken to place control areas in the exact same geologic regions and context as their associated intra-PSR areas. All craters that intersect the border of the counting area are mapped and included in the analysis. All cumulative CSFD curves have been plotted and analyzed using the CraterTools and CraterStats2 software package (e.g., ref. [66]), following the guidelines established by ref. [67].

## Data availability

The raw NAC EDR data used in this study are available via the LROC page at http://lroc.sese.asu.edu/data/. Auxiliary data, such as LOLA products, are available via the NASA PDS (https://pds.nasa.gov) and USGS pages (https://www.usgs.gov/centers/astrogeology-science-center/data-tools). HORUS images and its training dataset will be available on request in the near future. The processed and validated data generated in this study are provided in the supplementary information of this article.

## Code availability

We used open-access/open-source software to process and analyze the data used for this paper, specifically Python (https://www.python.org/), PyTorch[68] https://pytorch.org/, Isis3 (e.g., ref. [64], https://isis.astrogeology.usgs.gov/), and planetary image (https://pypi.org/project/planetaryimage/). We plan to open-source the HORUS code at a later date; all details are available here: https://openaccess.thecvf.com/content/CVPR2021/html/Moseley_Extreme_Low-Light_Environment-Driven_Image_Denoising_Over_Permanently_Shadowed_Lunar_Regions_CVPR_2021_paper.html.

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

## Acknowledgements

V.T.B. would like to thank Harald Hiesinger and Carolyn van der Bogert for the insightful discussions about crater counting in particularly small areas with unknown physical properties and Ernst Hauber for generously sharing images of permafrost-related features acquired on Spitsbergen, Norway. The initial development of HORUS was an output from the Frontier Development Lab; all authors would like to thank Loveneesh Rana for his contribution to all FDL-related work. All authors are grateful for the technical, financial, and mentorship support by Google Cloud, the Frontier Development Lab, the SETI Institute, and the Luxembourg Space Agency. All authors thank L. Silverberg for creating the HORUS mission patch. V.T.B. gratefully acknowledges the financial support by the International Max Planck Research School (IMPRS) at the Max Planck Institute for Solar System Research (MPS). This research has made use of the USGS Integrated Software for Imagers and Spectrometers (ISIS).

## Author contributions

V.T.B., B.M., and I.L.-F. were involved in the development of HORUS. V.T.B. and B.M. were involved in conceptualization, resources, data curation, writing, preparation, visualization, and validation of this work. V.T.B. and M.S. were involved with interpretation and investigation. V.T.B. was involved with formal analysis and revision. All authors were involved in the review of this work.

## Funding

## Competing interests

The authors declare no competing interests.
