## [Peer Review File · Nature Communications]

REVIEWER COMMENTS

Reviewer #1 (Remarks to the Author):

Summary

This article presents results from a machine learning technique to de-noise orbital images of permanently shadowed regions on the Moon. Multiple applications of this technique are explored including crater counting, trafficability and a search for signatures of ice that could be present.

Overall, the results are quite impressive and the work will have broad interest across the planetary science community and others. I do have some concerns that should be addressed before publication, as described below.

General Comments

1. The abstract states “enabling us to identify potential small-scale geomorphic evidence for a surficial high-porosity layer and near-surface ice, whilst not providing evidence for surface frost”

I didn't see any positive evidence or even indications for near-surface ice presented in the remainder of the work, and in fact the opposite seems to be true, that no ice is visible in any of the corrected NAC images, and geomorphologic features that could be due to ice are also detected in sunlit areas where ice cannot be present. I suggest revising this sentence to not give the impression that this work has detected near-surface ice, which is how it reads currently.

2. Given the significant decrease in performance with lower counts, I would like to see some discussion of how many or what fraction of the 200,00 NAC images over the poles these methods can be expected to provide a significant benefit for. Obviously, the images shown in the figures are the ones with the best results, but it would be useful to see some statistics to gauge the overall utility to the NAC dataset as a whole.

3. Some of the referencing is suspect throughout the work: for example, why cite Prem et al. 2015 as the one modern paper to go along with the canonical Watson 61 and Arnold 79 on line 24? Rubanenko et al. 2019 is cited alongside the LOLA remote sensing papers that used the brightness from LOLA, but this was a very different study that used depth/diameter of craters. Hayne 2015 is cited for Diviner but that paper is about LAMP detections of ice. Please go back and thoroughly check that the references are well chosen to support the text.

Line-specific Comments

Line 31: This statement that “and are only able to provide data and imagery within large PSRs (on a kilometer scale)” is not strictly true: LOLA, LAMP and M3 all have spatial resolutions $\ll 1$ km.

Line 47: Remove comma after “small”

Line 50: “Without high resolution data and imagery, PSRs are terra incognita, and this complicates mission planning and generates questions such as: Where can a given PSR be entered safely? Does it contain hazards to avoid? Where can we take samples or drill?”

I think this point is overstated. We have 5 m/px LOLA DEMs over the poles, and with recent advancements by Baker et al. 2021 these are very high-quality datasets from which hillshades can be generated and the geomorphology and trafficability can be analyzed. This fact should not be omitted here. Yes, the corrected NAC images are better, but we're not exactly blind without them.

Line 212: But do the PSR AoIs feature more larger craters than the sunlit comparisons? If not, then this may

not be the correct explanation.

Line 216: Is it physically realistic to maintain significant porosity to a depth of 1.4 m? This seems unlikely. The normal lunar regolith is ~40-50% porous in the upper 10-ish cm but this falls off quite quickly with depth. Are we talking about 70% porosity? 80%?

Line 336: "suggested that the presence of surface frost in PSRs could be mimicked by rocks and surface roughness". Radar cannot see a frost whether it is present or not. The works by Fa, Cai and Eke were focused on circular polarization ratios that were interpreted as thick pure ice layers near the surface by others (e.g., Spudis et al. 2013).

Reviewer #2 (Remarks to the Author):

This is a interesting, innovative and scholarly paper regarding the geology of regions of the Moon that are permanently shaded from sunlight and may contain significant amounts of ice. It applies a new image processing technique to reduce noise in existing imagery, then interprets the results. The interpretations are reasonable in the context of the imagery presented and well grounded in the literature.

However, the most critical results depend upon the detection or non-detection of features (craters, boulder tracks) that are near the scale at which the imagery begins to feature false negatives as shown and discussed in the paper. The analysis lacks controls that are essential to assessing the validity of the results. For example, the authors could use imagery of equatorial regions that have data obtained while illuminated, and also in shadow due to the lunar day/night cycle, to demonstrate that anomalously low small crater counts are not an artifact of the image processing technique. Furthermore, illumination within shadowed areas arises from nearby sunlit terrain that can have a very large solid angle with respect to the Sun, and have an important effect on the nature of shadows, compared to the narrow angle illumination of the Sun. The same shortcoming applies to conclusions arising from boulder tracks. These are abundant Moonwide, so statistically significant controlled experiments could be applied. The non-detection of lobate features is similarly suspect given the illumination differences, and the lack of a clearly established influence of the image processing method on analogous features.

A lesser point is that the crater size frequency data are presented in a non-standard manner, and in a way that results in high sensitivity to small crater undercounting. The authors may want to conform to the standards given in:

Crater Analysis Techniques Working Group. (1979). Standard techniques for presentation and analysis of crater size-frequency data. *Icarus*, 37(2), 467-474.

In summary, the conclusions of the paper are quite important, but the controls on the experiment are not sufficient to judge the validity of these conclusions.

Paul Lucey

Response to Reviewers

Dear reviewers, dear Sir or Madam,

Thank you very much for **your comments and suggestions (black)**. We carefully considered and **addressed every single comment here (green)** and in the revised manuscript. Changes in the revised manuscript are marked as **CHANGE**.

We look forward to receiving further feedback from you soon. Thank you very much and best regards,

Valentin Bickel (on behalf of all authors)

General changes

Added: added new references (Lemelin et al., 2021), (Crater Analysis Techniques Working Group, 1979), (Riedel et al., 2018), (Reiss et al., 2021)

Added: updated selected figures & captions

Added: updated specific sections of the main paper, methods, acknowledgements, and supplementary information

Removed: removed the following unused references (Haruyama et al., 2013), (Prem et al., 2015), (Florensky et al., 1978), (Carrier et al., 1991)

Reviewer #1

Summary

This article presents results from a machine learning technique to de-noise orbital images of permanently shadowed regions on the Moon. Multiple applications of this technique are explored including crater counting, trafficability and a search for signatures of ice that could be present.

Overall, the results are quite impressive and the work will have broad interest across the planetary science community and others. I do have some concerns that should be addressed before publication, as described below.

Our reply: Dear Sir or Madam, thank you very much for your productive feedback and for sharing your concerns. We addressed all points in great detail below and throughout the manuscript.

General Comments

1. The abstract states “enabling us to identify potential small-scale geomorphic evidence for a surficial high-porosity layer and near-surface ice, whilst not providing evidence for surface frost”

I didn't see any positive evidence or even indications for near-surface ice presented in the remainder of the work, and in fact the opposite seems to be true, that no ice is visible in any of the corrected NAC images, and geomorphologic features that could be due to ice are also detected in sunlit areas where ice cannot be present. I suggest revising this sentence to not give the impression that this work has detected near-surface ice, which is how it reads currently.

Our reply: Thank you for pointing this out. We adapted the sentence to avoid giving a wrong impression. The modified sentence is:

“We show that our method is able to reveal previously unseen geomorphological features such as boulders and craters down to 3 meters in size, whilst not finding evidence for surface frost or near-surface ice.”

2. Given the significant decrease in performance with lower counts, I would like to see some discussion of how many or what fraction of the 200,00 NAC images over the poles these methods can be expected to provide a significant benefit for. Obviously, the images shown in the figures are the ones with the best results, but it would be useful to see some statistics to gauge the overall utility to the NAC dataset as a whole.

Our reply: Thank you for this comment, we agree that a short discussion of this would add value to the paper. The number of NAC images that benefit from HORUS post-processing varies from PSR to PSR, as the secondary illumination is mainly controlled by the local topography. For the areas of interest we have been looking at so far, we observe that ~25% of all NAC images benefit from HORUS post-processing, on average. Extrapolating this percentage, roughly ~50,000 of the ~200,000 polar NAC images might benefit from HORUS post-processing, enabling a plethora of PSR-related science studies. We will further assess this in future work.

We added a brief discussion of this to the Discussion section of the manuscript:

“Importantly, HORUS allows us to tap into the large and growing archive of more than 200,000 NAC images over the lunar north and south pole (beyond 80°N and S). We observe that roughly 25% of the NAC images covering the Aols considered for this work had useable signal allowing HORUS post-processing to be used, meaning that there could be an estimated total of ~50,000 HORUS images available, enabling studies of an extensive number of small and large shadowed regions, including studies of the temporal and seasonal variations of TSRs and PSRs at unprecedented temporal and spatial resolution, i.e., on timescales from days to years (Figure 4).”

3. Some of the referencing is suspect throughout the work: for example, why cite Prem et al. 2015 as

the one modern paper to go along with the canonical Watson 61 and Arnold 79 on line 24? Rubanenko et al. 2019 is cited alongside the LOLA remote sensing papers that used the brightness from LOLA, but this was a very different study that used depth/diameter of craters. Hayne 2015 is cited for Diviner but that paper is about LAMP detections of ice. Please go back and thoroughly check that the references are well chosen to support the text.

Our reply: Thank you for pointing this out. We carefully checked the manuscript and added or removed references, where required. For example, we replaced Prem et al. (2015) with the more appropriate Feldman et al. (2001) – ‘Evidence for water ice near the lunar poles’ – and added the very recent Lemelin et al. (2021), too.

Regarding LOLA and Rubanenko et al. (2019): In order to avoid potential confusion, we reworded the respective section as follows:

“Evidence for volatiles on or underneath the surface of lunar cold traps, involving varying kinds of modeling and associated assumptions, has been found by the LCROSS impactor mission (Colaprete et al., 2010; Schultz et al., 2010), the Lunar Orbiter Laser Altimeter (LOLA, Fisher et al., 2017; Qiao et al., 2019; Rubanenko et al., 2019), the Diviner Lunar Radiometer Experiment (DLRE, Hayne et al., 2015; Hayne et al., 2017), the Lyman Alpha Mapping Project (LAMP, Gladstone et al., 2012), and the Moon Mineralogy Mapper (M3, Li et al., 2018).”

Regarding Hayne et al. (2015): you are right, this paper is not using Diviner data exclusively, but in combination with LAMP data. Despite that, we think it is best associated with Diviner (we have to cite it either for Diviner or LAMP).

Line-specific Comments

Line 31: This statement that “and are only able to provide data and imagery within large PSRs (on a kilometer scale)” is not strictly true: LOLA, LAMP and M3 all have spatial resolutions \ll 1 km.

Our reply: Thank you for pointing this out. We added the term “sub-kilometer” to reflect that fact:

“However, the scientific instruments used by these studies are limited by their signal-to-noise ratio and/or resolution over PSRs, and are only able to provide data and imagery within large PSRs (on a kilometer to sub-kilometer scale).”

Line 47: Remove comma after “small”

Our reply: Thank you for pointing this out, we removed the comma.

Line 50: “Without high resolution data and imagery, PSRs are terra incognita, and this complicates mission planning and generates questions such as: Where can a given PSR be entered safely? Does it contain hazards to avoid? Where can we take samples or drill?”

I think this point is overstated. We have 5 m/px LOLA DEMs over the poles, and with recent advancements by Baker et al. 2021 these are very high-quality datasets from which hillshades can be generated and the geomorphology and trafficability can be analyzed. This fact should not be omitted here. Yes, the corrected NAC images are better, but we’re not exactly blind without them.

Our reply: Thank you for bringing this up. We modified the sentence to better reflect the fact that there are other datasets that enable geomorphological analyses of PSRs:

“Without high resolution data and imagery (< 5 m/pixel), the meter-scale geomorphology of PSRs remains unknown, and this complicates mission planning and generates questions such as: Where can a given PSR be entered safely? Does it contain hazards to avoid? Where can we take samples or drill?”

Line 212: But do the PSR Aols feature more larger craters than the sunlit comparisons? If not, then this may not be the correct explanation.

Our reply: Thank you for sharing this concern. Following a suggestion from reviewer #2 we implemented the CraterTools and CraterStats2 software packages to conform to CSFD plot standards. In addition, we ran two new, CSFD-specific experiments to specifically test whether the originally observed CSFD shift has a physical cause. We found that the shift was most likely caused by an unexpected observational bias introduced by HORUS (craters with diameters $< \sim 8.5$ m are not always resolved) that was additionally amplified by the inappropriate style of the original Figure 6.

This means that, unfortunately, we are not able to present evidence for or against a shallow high porosity layer in lunar PSRs. We updated the entire manuscript and supplementary information accordingly.

Line 216: Is it physically realistic to maintain significant porosity to a depth of 1.4 m? This seems unlikely. The normal lunar regolith is ~ 40 - 50% porous in the upper 10-ish cm but this falls off quite quickly with depth. Are we talking about 70% porosity? 80%?

Our reply: Thank you for sharing this concern. Before we identified the observational bias of HORUS, we considered e.g. volatiles and/or ice to be potentially responsible for high porosity values at greater depth. Due to HORUS's observational bias, we are not able to keep a discussion of regolith porosity in the manuscript, unfortunately (see reply above). We hope that future missions (such as VIPER) provide the much-needed ground-truth to answer this important question.

Line 336: "suggested that the presence of surface frost in PSRs could be mimicked by rocks and surface roughness". Radar cannot see a frost whether it is present or not. The works by Fa, Cai and Eke were focused on circular polarization ratios that were interpreted as thick pure ice layers near the surface by others (e.g., Spudis et al. 2013).

Our reply: Thank you for pointing this out; the original sentence mixed up several aspects that should not be mixed. We modified the sentence in order to improve its clarity:

"Work based on Mini-RF Radar data (Fa & Cai, 2013; Fe & Eke, 2018) suggested that the presence of surface frost and buried ice deposits in PSRs could be mimicked by surface roughness and rocks (skree), however, our observations made from HORUS images in small PSRs indicate that the respective rocks – if present – would need to be completely buried or much smaller than ~ 3 m in diameter, at least in the studied small lunar PSRs."

Reviewer #2

This is an interesting, innovative and scholarly paper regarding the geology of regions of the Moon that are permanently shaded from sunlight and may contain significant amounts of ice. It applies a new image processing technique to reduce noise in existing imagery, then interprets the results. The interpretations are reasonable in the context of the imagery presented and well grounded in the literature.

Our reply: Dear Prof. Lucey, thank you very much for your productive feedback. We made substantial changes to the manuscript in response to your review. A detailed reply to your concerns is shown below.

However, the most critical results depend upon the detection or non-detection of features (craters, boulder tracks) that are near the scale at which the imagery begins to feature false negatives as shown and discussed in the paper. The analysis lacks controls that are essential to assessing the validity of the results. For example, the authors could use imagery of equatorial regions that have data obtained while illuminated, and also in shadow due to the lunar day/night cycle, to demonstrate that anomalously low small crater counts are not an artifact of the image processing technique. Furthermore, illumination within shadowed areas arises from nearby sunlit terrain that can have a very large solid angle with respect to the Sun, and have an important effect on the nature of shadows, compared to the narrow angle illumination of the Sun. The same shortcoming applies to conclusions arising from boulder tracks. These are abundant Moonwide, so statistically significant controlled experiments could be applied. The non-detection of lobate features is similarly suspect given the illumination differences, and the lack of a clearly established influence of the image processing method on analogous features.

Our reply: We agree that the thorough validation of the produced and used HORUS images is essential. In your feedback, we identify two major concerns: A) The observation that HORUS tends to produce false-negatives for small features below a minimum feature size which depends on the signal-to-noise ratio, potentially affecting the derived CSFD curves; and B) the different nature of secondary illumination /shadows within PSRs impede correct interpretations of geomorphology. Following your suggestions, we implemented three additional, application-specific control tests/experiments to address these two major concerns and strengthen the scientific robustness of this paper:

Application 1) Boulder detection & diameters: As a control test, we map the distribution of boulders ($n = 5$) within a TSR in sunlit (regular NAC) and shadowed conditions (HORUS image) and compare their diameters as measured in the different images (both images have a similar spatial resolution). The small number of boulders is governed by the scarcity of boulders across the studied regions. We observe that boulders appear only slightly larger in the analyzed HORUS image (shadowed conditions), ~ 0.9 m larger on average (~ 11 % larger relative to sunlit conditions), which is less than one pixel. These results indicate that there might be a subtle observational bias introduced by HORUS images for this particular application. We added this test and insight to the manuscript:

"We perform a test within a TSR that is located right next to A004 PSR 3, mapping boulders ($n = 5$) in sunlit (regular NAC image) and shadowed conditions (HORUS image). The used HORUS image is the same as used for the intra-PSR boulder count of A004 PSR 3. The small number of boulders is governed by the overall scarcity of boulders in the studied TSRs. We observe that boulders that are observable in sunlit and shadowed conditions appear slightly larger in the considered HORUS image (shadowed conditions), roughly ~ 0.9 m larger on average (~ 11 % larger relative to boulders observed in sunlit conditions), which is less than one pixel, meaning that HORUS might potentially introduce a subtle observational bias, at least for small, high intensity features such as boulders, depending on the DN and SNR of the respective HORUS image."

Application 2) Crater mapping & CSFD curves: As a control test, we perform two additional tests in a TSR, as you suggested.

In test #1, we map the distribution of craters ($n = 60$) within a TSR in sunlit (regular NAC) and shadowed conditions (HORUS image) and compare the CSFD curves as measured in the

different images (both images have a similar spatial resolution). We exclusively consider craters for this test that are visible in both, sunlit and shadowed images – the idea is to check whether crater diameter measurements are affected by HORUS. Test #1 shows that the derived CSFD curves (sunlit – shadowed) are nearly identical (please see new Figure 6), meaning that HORUS does not significantly affect the measurement accuracy (<< one pixel) (new Figure 6).

In test #2, we map the distribution of all visible craters within a TSR in sunlit (regular NAC) and shadowed conditions (HORUS image) and compare the CSFD curves as measured in the different images (both images have a similar spatial resolution). This time we map all craters visible within the area in a given image – the idea is to check whether HORUS introduces an observational bias, i.e., resolves the crater population in the same way as a sunlit image. Test #2 suggests that not all of the craters with diameters <~8.5 m are resolved by HORUS, indicating that there is an observational bias (new Figure 6).

This means that the original observation of a lack of very small craters in PSRs as well as the potential evidence for a high porosity regolith layer are not valid anymore, unfortunately. We updated the entire manuscript and supplementary information accordingly. We describe the tests and their results in the manuscript:

"We perform an additional validation test within a TSR that is located within A004, in between PSRs 1, 2, and 3 (89.82°S, 44.01°W). We map all craters in a pre-defined area in sunlit (regular NAC CDR image) and shadowed conditions (the same HORUS image used for the intra-PSR CSFD curve) and compute various CSFDs (shown in Figure 6). We draw two conclusions from this test: the first is that for craters which are observable in both images, the crater diameter measurements performed in HORUS images do not suffer from any observation bias; in other words, HORUS does not appear to increase the diameter of craters compared to the sunlit image (Figure 6 left). The second is that HORUS does not reliably resolve a substantial fraction of the crater population with diameters smaller than ~8.5 m; in other words, HORUS contains some, but not all craters <~8.5 m in diameter (Figure 6 right); we note that the observed threshold diameter could be subject to change, depending on the DN and SNR of the used images. This leads us to discard hypotheses (A) and (B), although we acknowledge the large statistical uncertainties associated with crater counting in such small Aols, which is a physical limitation of the study of small PSRs, however. The analysis of additional, small PSRs will help to gather more confident statistics."

Application 3) Surface texture & lobate feature mapping: As a control test, we analyze the visual appearance of surface texture and lobate-type features within a TSR in sunlit (regular NAC) and shadowed conditions (HORUS image) (both images have a similar spatial resolution). We observe that – overall - HORUS images appropriately display the surface texture as observed in sunlit conditions (not only craters and boulders, but also ridges, overall texture, lobes, etc.). Thus, it appears HORUS is able to detect such features. We added an illustrative figure to the supplementary information section (Fig. S23).

We note that varying illumination conditions might highlight different aspects of the surface texture present in a PSR, as is the case for sunlit regions as well. In this study we also use multi-frame image validation, which allows further verification by requiring features are consistent across the frames, which typically have varying illumination conditions (as performed for each AOI analyzed in this study). We point out that the image validation methods as presented in Figures 3 and 4 allow for the effective verification of surface texture and features in general. We added these insights to the Subsurface Ice section:

"We perform a qualitative test within a TSR that is located close to A001 PSR 10 (89.59°S, 118.73°W), comparing the visual appearance of surface texture and lobate-type features in sunlit (regular NAC image) and shadowed conditions (HORUS image). We observe that – overall – the analyzed HORUS images appropriately display the surface texture and relevant features as observed in sunlit conditions (Fig. S23). However, we note that varying illumination conditions might highlight different aspects of the surface texture present in a PSR, as is the case for sunlit regions as well. In this study we also use multi-frame image validation, which allows further verification by requiring features are consistent across the frames, which typically have varying illumination conditions (see Figures 3 and 4)."

In order to better represent and communicate the remaining uncertainty related to these three additional, application-specific experiments, we updated the overall tone of the manuscript. We intend to communicate that “if we can see a feature in multiple HORUS images, it’s likely there”, while trying to avoid the incorrect impression that “if we cannot see a feature in a HORUS image, it’s not there”. For example, in the Discussion we added:

“Furthermore, our validation suggests that whilst HORUS is able to resolve small features in PSRs, depending on the SNR available it may not be able to fully resolve all small features present.”

A lesser point is that the crater size frequency data are presented in a non-standard manner, and in a way that results in high sensitivity to small crater undercounting. The authors may want to conform to the standards given in:

Crater Analysis Techniques Working Group. (1979). Standard techniques for presentation and analysis of crater size-frequency data. *Icarus*, 37(2), 467-474.

Our reply: Thank you for pointing this out, we carefully read the working group report. We used CraterTools and CraterStats2 (a commonly used crater analysis software package) to ensure “good practice” for the revised version of Fig. 6. We additionally used these tools to produce a new supplementary figure (Figure S22). The updated figures in combination with the two additional crater counting-related tests helped to reveal an observational bias of HORUS, as discussed above.

In summary, the conclusions of the paper are quite important, but the controls on the experiment are not sufficient to judge the validity of these conclusions.

Paul Lucey

Our reply: Thank you for reviewing this work. We hope our revision and additional experiments are sufficient to meet your concerns.

REVIEWERS' COMMENTS

Reviewer #1 (Remarks to the Author):

Line 39: Change "dominates" to "dominate".

Line 74: There is a special character for much less than: << instead of two less thans: <<.

Line 94: Use consistent spacing between values and units (c.f., Line 93).

Lines 122-124 and throughout: Use "i.e.," instead of "i.e.". Same for e.g. throughout.

Line 252: "West" should not be capitalized.

Line 356: Suggest removing "mineralogy", I don't see how HORUS could be used to say much if anything about mineralogy which is expected to be almost homogeneous over the polar regions that are all highlands terrain.

Reviewer #2 (Remarks to the Author):

I reviewed the manuscript and am satisfied with the response and recommend publication.

Response to Reviewers

Dear reviewers, dear Sir or Madam,

Thank you very much for **your final comments and suggestions (black)**. We carefully considered and **addressed every single comment here (green)** and in the revised manuscript. Original changes in the revised manuscript are marked as CHANGE, the latest changes are marked as CHANGE.

Thank you very much and best regards,

Valentin Bickel (on behalf of all authors)

General changes

Added: A small number of minor changes to the title, abstract, figures, footnotes, main text, methods, and SUP, following Nature Communications editorial requests.

Reviewer #1

Our reply: Dear Sir or Madam, thank you very much for your final set of feedback and comments. We implemented all of them throughout the manuscript.

Line 39: Change “dominates” to “dominate”.

Line 74: There is a special character for much less than: << instead of two less thans: <<.

Line 94: Use consistent spacing between values and units (c.f., Line 93).

Lines 122-124 and throughout: Use “i.e.,” instead of “i.e.”. Same for e.g. throughout.

Line 252: “West” should not be capitalized.

Our reply: Thank you, we implemented all suggestions.

Line 356: Suggest removing “mineralogy”, I don’t see how HORUS could be used to say much if anything about mineralogy which is expected to be almost homogeneous over the polar regions that are all highlands terrain.

Our reply: Thank you for this suggestion; we removed the term “mineralogy” from the respective paragraph.

Reviewer #2

I reviewed the manuscript and am satisfied with the response and recommend publication.

Our reply: Dear Prof. Lucey, thank you very much for your review, feedback, and recommendation!